# Snail-Overexpression Induces Epithelial-mesenchymal Transition and Metabolic Reprogramming in Human Pancreatic Ductal Adenocarcinoma and Non-tumorigenic Ductal Cells

**DOI:** 10.3390/jcm8060822

**Published:** 2019-06-08

**Authors:** Menghan Liu, Sarah E. Hancock, Ghazal Sultani, Brendan P. Wilkins, Eileen Ding, Brenna Osborne, Lake-Ee Quek, Nigel Turner

**Affiliations:** 1Department of Pharmacology, School of Medical Sciences, University of New South Wales, Sydney, 2052 NSW, Australia; mliu6171@uni.sydney.unsw.edu.au (M.L.); sarah.hancock@unsw.edu.au (S.E.H.); ghazal.sultani@unsw.edu.au (G.S.); b.wilkins@victorchang.edu.au (B.P.W.); e.ding@unsw.edu.au (E.D.); b.osborne@unsw.edu.au (B.O.); lake-ee.quek@sydney.edu.au (L.-E.Q.); 2Charles Perkins Centre, School of Mathematics and Statistics, The University of Sydney, Sydney, 2006 NSW, Australia

**Keywords:** SNA1, metabolomics, glucose metabolism, tumor metabolism, epithelial-mesenchymal transition, pancreatic adenocarcinoma

## Abstract

The zinc finger transcription factor Snail is a known effector of epithelial-to-mesenchymal transition (EMT), a process that underlies the enhanced invasiveness and chemoresistance of common to cancerous cells. Induction of Snail-driven EMT has also been shown to drive a range of pro-survival metabolic adaptations in different cancers. In the present study, we sought to determine the specific role that Snail has in driving EMT and adaptive metabolic programming in pancreatic ductal adenocarcinoma (PDAC) by overexpressing Snail in a PDAC cell line, Panc1, and in immortalized, non-tumorigenic human pancreatic ductal epithelial (HPDE) cells. Snail overexpression was able to induce EMT in both pancreatic cell lines through suppression of epithelial markers and upregulation of mesenchymal markers alongside changes in cell morphology and enhanced migratory capacity. Snail-overexpressed pancreatic cells additionally displayed increased glucose uptake and lactate production with concomitant reduction in oxidative metabolism measurements. Snail overexpression reduced maximal respiration in both Panc1 and HPDE cells, with further reductions seen in ATP production, spare respiratory capacity and non-mitochondrial respiration in Snail overexpressing Panc1 cells. Accordingly, lower expression of mitochondrial electron transport chain proteins was observed with Snail overexpression, particularly within Panc1 cells. Modelling of ^13^C metabolite flux within both cell lines revealed decreased carbon flux from glucose in the TCA cycle in snai1-overexpressing Panc1 cells only. This work further highlights the role that Snail plays in EMT and demonstrates its specific effects on metabolic reprogramming of glucose metabolism in PDAC.

## 1. Introduction

Originating from the ductal cells of exocrine pancreas, pancreatic ductal adenocarcinoma (PDAC) is arguably the most lethal type of common cancer, and its dismal prognosis has remained relatively unchanged over the past three decades [1,2]. Decades of intensive research and clinical investigation have yielded a wealth of knowledge of pancreatic cancer pathophysiology, but effective treatment strategies are still in urgent demand to battle against the rise in pancreatic cancer-related mortalities [2]. The main causes of PDAC-related mortality are the frequent occurrence of metastatic spread and resistance to currently available therapeutic interventions, both of which are partially underlined by a complex process termed the epithelial-mesenchymal transition or EMT [3,4]. The cellular transition from epithelial to mesenchymal phenotype involves profound changes in gene expression patterns, which impart cells with a series of functional properties such as increased migratory potential, invasiveness, resistance to apoptotic stimuli and stemness [5]. In response to EMT-inducing signals commonly existing in the tumor micro-environment, a network of intracellular pathways is activated to convey the message to the EMT executioners for transcriptional regulation [6]. 

The zinc finger transcription factor Snail was the first identified and is the best characterized EMT effector, and it primarily controls EMT via repressing E-cadherin expression [7,8]. To orchestrate the EMT process, Snail is able to upregulate the mesenchymal genes N-cadherin, vimentin, fibronectin, the matrix metalloproteases (MMPs) and other EMT-inducing transcription factors including Twist1, Zeb1 and Zeb2 [7,9]. In PDAC, the functional significance of Snail-induced EMT has been exemplified by observations of clinical samples and experimental manipulations. Immunohistochemical staining of PDAC surgical specimens has revealed strong Snail expression in 35–80% of samples, which was tightly associated with lymph node invasion and distant metastasis [10,11]. Highly metastatic PDAC cell sublines have also been reported to possess EMT-like phenotype and Snail upregulation when compared with the bulk of tumor cells [12]. Such observations are confirmed by studies using cultured cells where Snail overexpression in Panc1, AsPC-1 and BxPC-3 cells led to overt EMT with alterations in morphology and gene expression and increased transwell invasion capacity [13,14,15]. Conversely, experimental knock-down of Snail in PDAC cell lines results in increased E-cadherin expression and translocation to the membrane and reduced tumorigenicity [16]. 

Over the past decade, metabolic reprogramming has been recognized as a hallmark in oncogenic transformation in PDAC and other cancers [17,18,19]. The well-known aerobic glycolysis or Warburg effect (upregulated glucose uptake and lactate production) has been shown to confer proliferative and survival advantages in multiple cancer types by supplying sufficient bioenergetic precursors and NADPH [20]. PDAC cells also display elevated glutaminolysis to maintain redox balance and scavenge extracellular fatty acids/amino acids to survive in a hostile microenvironment with limited fuel supply [18,21,22,23]. Heterogeneity exists between metabolic profiles of different cancer types and between different cancer cell populations within the same tumor, owing to the context-specific oncogenic signaling events and micro-environmental factors present [24,25]. The process of EMT involves major changes to the gene expression network and cellular phenotype. It is, therefore, likely to be accompanied by metabolic alterations to accommodate the shift in cell’s priority from proliferation to invasion of neighboring tissues and to adapt to changes in the environment. Indeed, a wide range of metabolic alterations have been observed with induction of EMT status in breast, lung, ovarian, cervical and prostate cancers, although the nature of the metabolic reprogramming varies widely across the studies [26,27,28,29,30,31,32,33,34,35,36,37,38,39,40]. We have previously shown augmentations of glucose consumption and lactate output in Panc1 cells undergoing tumor necrosis factor-α (TNFα)- and transformation growth factor-β (TGFβ)-induced EMT, with differential molecular changes observed in the two models [41].

Alongside changes to cell metabolism in our previous study [41], we also observed the induction of Snail expression concurrently with change in EMT status in Panc1 cell upon treatment with TGFβ or TGFβ combined TNFα. In view of the importance of Snail-dependent EMT in underlying PDAC-related lethality, we sought to induce EMT in Panc1 and the non-tumorigenic human pancreatic ductal epithelial (HPDE) cells [42] via Snail overexpression to investigate the metabolic consequences. Specifically, we chose to compare the effects of Snail overexpression in a pancreatic cell line already on the EMT spectrum (Panc1) to that of a purely epithelial pancreatic cell line (HPDE) to study the specific consequences of Snail induction at different points across the EMT differentiation spectrum. Here we report that EMT in both cell lines is associated with elevated glucose uptake and lactate excretion, as well as downregulation of proteins in the mitochondrial electron transport chain (ETC).

## 2. Experimental Section

### 2.1. Antibodies and Reagents

Antibodies used are listed as follows: Snail (3879), Vimentin (5741), LDH-A (2012, Cell Signaling Technology, Danvers, MA, USA); E-cadherin (sc-21791), N-cadherin (sc-7939), Beta-actin (sc-47778, Santa Cruz, TX, USA); Hexokinase II (ab37593), Total OXPHOS Human WB Antibody Cocktail (ab110411, Abcam, Cambridge, UK). All other reagents were from Sigma-Aldrich (Sydney, NSW, Australia) unless stated otherwise.

### 2.2. Cell Culture 

Panc1 cells from ATCC were cultured in Dulbecco’s modified Eagle’s medium (DMEM) containing 4.5 g/L glucose supplemented with 10% fetal bovine serum and penicillin-streptomycin. The human pancreatic ductal epithelial (HPDE) cell line [42] were a kind gift from Dr. Phoebe Phillips at Lowy Cancer Institute, UNSW Australia. HPDE cells were cultured in keratinocyte serum-free (KSF) media (ThermoFisher Scientific, Waltham, MA, USA) containing 1.6 g/L glucose supplemented with epidermal growth factor (5 ng/mL), bovine pituitary extract (50 ug/mL) and penicillin-streptomycin.

### 2.3. Overexpression of the SNAI1 Gene

Plasmid containing human SNAI1 (encoding Snail) cDNA (Addgene plasmid 23347) was a kind gift from Bob Weinberg [43]. Stable SNAI1-expressing Panc1 cell line was generated using retroviral-mediated pBabe-puro-Snail infection. Briefly, 8 × 10^5^ Hek293-FT cells were seeded on 10 cm culture dishes and allowed to attach overnight before being transiently transfected with gag/pol and VSV-G packaging plasmids, along with the pBabe-puro-Snail plasmid or empty pBabe-puro vector. The culture media for Hek293-FT was refreshed 12 hours later and media containing viral particles were harvested 24 hours and 48 hours after media refreshment. The 24 hours and 48 hours media was combined and applied to target cells at 40% confluence with polybrene (8 μg/μL) for 24 hours. A pooled cell population was used for experiments following puromycin selection. 

### 2.4. Cell Morphology

Pictures of cells were taken using a phase contrast microscope (Nikon Eclipse TS100, Nikon, Tokyo, Japan) attached to a camera (Nikon digital sight) under 40× or 100× magnification.

### 2.5. Western Blotting

Cells (1 × 10^6^) growing on 6-well plates were lysed in RIPA buffer containing protease inhibitors as described previously [44]. After denaturation, samples (20 ug protein) were resolved by 10% polyacrylamide gel electrophoresis (187 V, 1 h) and transferred to PVDF membranes (65 V, 65 min). For immunoblotting, membranes were blocked in 5% skim milk in tris-buffered saline containing Tween 20 (TBST), incubated with primary antibodies overnight at 4°C, washed with TBST and incubated with secondary antibodies in 5% skim milk in TBST for an hour. After washing, membranes were developed with enhanced chemiluminescence reagents (Western Lighting Plus-ECL, Perkin Elmer, Waltham, MA, USA) and visualized under Las4000 imager (GE Healthcare, Chicage, IL, USA). Densitometry was performed using ImageJ software by obtaining the optical density of each band.

### 2.6. Quantitative PCR

Total RNA was extracted using the Roche High Pure RNA Extraction kit according to manufacturer’s instructions (Roche, Basel, Switzerland). RNA (1 μg) was then reverse-transcribed using the Roche Transcriptor first strand cDNA synthesis kit. The resulting cDNA was mixed with primers (primer sequences sourced from Sigma KiCqstart or the Primer Bank [45]) and SYBR-Green (Roche, Basel, Switzerland) in 96-well plates. Quantitative PCR was performed using Roche 480 Light Cycler to obtain Ct values for each gene of interest and a housekeeper beta-actin. Analysis was conducted using the ΔΔCT method. 

### 2.7. Wound Healing Assay

Cells were grown in 6-well plates and scratch wounds were created by scraping confluent cell monolayers with a sterile pipette tip on 3 sites on each well. The cells were then incubated under normal conditions with refreshed media under the Nikon Tie inverted time-lapse microscope for 24 hours. Migration at 24 hours was quantified by measuring the area closed between two moving borders of the cells from each scratch. Values from the 3 wounds on each of the triplicate wells were averaged and 3 independent experiments were carried out.

### 2.8. Measurement of Cell Proliferation

Cells were seeded in 6-well plates at a density of 1.5x10^5^ per well. After 4 days, cells were rinsed with PBS and immersed in 0.5% crystal violet (w/v)/50% methanol (v/v) solution and left to fix for 20 min. After fixation, cells were gently rinsed to remove all the crystal violet solution and allowed to dry overnight. The next day, fixed cells were solubilized with 1% SDS at 37°C and 50 μL of the solution was taken for absorbance measurements at 570 nm using a plate reader as an indication of cell number of the well. 

### 2.9. Glucose Uptake Assay

Glucose uptake was assessed using the glucose analogue 2-deoxyglucose (2-DG). Cytochalasin B (25 μM) was applied to control wells for 15 min before the assay and during the assay to give a measure of background glucose uptake.

After washing with PBS, cells in 6-well plates were incubated in Ringer solution (140 mM NaCl, 20 mM HEPES, 5 mM KCl, 2.5 mM MgSO4, 1.2 mM CaCl2, pH7.4) containing 10 μM 2-DG and 0.5 μCi/mL radio-labelled ^3^H-2-DG for exactly 8 min. Following incubation, cells were washed with cold PBS and lysed in 1 M NaOH. The amount of ^3^H radioactivity in lysates was counted using a beta-counter (Tri-Carb liquid scintillation counter, Perkin Elmer, Waltham, MA, USA) from which background was subtracted. Protein concentrations of lysates were measured using BCA assay (Pierce BCA protein assay kit, Thermo Fisher Scientific, Waltham, MA, USA) for normalization.

### 2.10. Lactate Assay

Lactate concentrations in cell culture media (72 hour after plating) were determined in a reaction mixture containing hydrazine hydrate (0.4 M, pH 9.0), EDTA (10 mM, pH 9.0) and NAD^+^ (0.5 mM). Samples and standards were added into 96-well plates followed by lactate dehydrogenase (10 units/well) and the amount of lactate was assessed by measuring the amount of NADH formed at 340 nm after 2 hours of incubation at 37°C (a timepoint when lactate conversion is complete). 

### 2.11. Bioenergetic Profiling of Oxygen Consumption and Extracellular Acidification in Snail Overexpressing and Control Cells

Cells were seeded at 2 × 10^4^ per well on a XF96 seahorse cell plate (Agilent Technologies, Santa Clara, CA, USA) in their respective growth media. The next day, cells were washed with Seahorse assay media containing 25 mM glucose, 2 mM L-glutamine, 1 mM sodium pyruvate, pH 7.4 equilibrated in the same media at 37 °C for 30 min in a CO_2_-free incubator. Bioenergetic profiling was performed by monitoring oxygen consumption and extracellular acidification rates at basal levels, followed by sequential injections of 1 μM oligomycin (an ATP synthase inhibitor), 0.5 μM Carbonyl cyanide-4-(trifluoromethoxy)-phenylhydrazone (FCCP) (a mitochondrial uncoupler) and 1 μM rotenone (a Complex I inhibitor) using the Seahorse XF96 Analyzer. The time-course of energetic profiles, as well as basal oxygen consumption, basal extracellular acidification rate and maximal oxygen consumption were calculated from the primary data. 

### 2.12. Glucose Oxidation

Glucose oxidation was measured in cells seeded in 6-well plates (1 × 10^6^ cells per well). Briefly, cells were washed with PBS and incubated in DMEM containing 1 g/L D-glucose and 2 μCi/ml ^14^C-glucose for 1 hour at 37 °C. After incubation, the culture media was added to 1 M perchloric acid and the CO_2_ released was absorbed in 1 M NaOH solution over 2 hours. The CO_2_ produced was quantified by counting ^14^C content in the NaOH solution using a beta-counter (Tri-Carb liquid scintillation counter, Perkin Elmer, Waltham, MA, USA).

### 2.13. Measurement of Half-Maximal Inhibitory Concentration 

The half-maximal inhibitory concentration (IC_50_) was measured by crystal violet assay. Cells were seeded at 1 × 10^4^ cells per well into a 96-well plate in triplicate and allowed to attach overnight. The next day cells were treated either with serial dilutions of gemcitabine or paclitaxel. The IC_50_ was measured after 48 hours by crystal violet assay as described above (*n* = 3 biological replicates), with cell viability being expressed relative to vehicle control (phosphate buffered saline for gemcitabine, 0.1% ethanol for paclitaxel). The IC_50_ was then calculated by non-linear regression by fitting the log-transformed drug concentration against relative cell viability.

For comparison under different glucose conditions, cells were allowed to adhere overnight in high glucose DMEM (i.e., 4.5 g/L glucose) before being treated with serial dilutions of gemcitabine spiked with an IC_75_ dose of paclitaxel in media containing either high or no glucose. 

### 2.14.^13^C metabolic Tracer Experiment and Metabolomics

Triplicates of Panc1 and HPDE cells were cultured in 6-well plates in their respective glucose-free DMEM and KSF media as described earlier. Approximately 4.5 g/L and 2.9 g/L of uniformly labelled ^13^C_6_-glucose was added to DMEM and KSF media respectively and cells were cultured for 5 hours. To measure the accumulation and ^13^C enrichment of extracellular pyruvate and lactate, 50 μL culture media was harvested hourly. The collected media were centrifuged (300 *x g*, 4°C) for 5 min and the supernatant stored at −30 °C until analysis by gas chromatography mass spectrometry (GCMS) using an extraction and derivatization described previously [46]. To measure ^13^C enrichment of intracellular metabolites, cells were quenched at the end of the 5-hour culture, and metabolites were then extracted and derivatized for GCMS analysis [41]. GCMS of derivatized metabolites was conducted using g a HP-5ms capillary column (0.25 mm i.d. × 30 m × 0.25 μm; Agilent J&W, Agilent Technologies, Santa Clara, CA, USA) installed in an Agilent HP 6890-5973 gas chromatography/mass selective detector.

### 2.14. ^13^C Flux Analysis

Flux modelling was performed to explain the activity of catabolic pathways used by Panc1 and HPDE to metabolize glucose. Metabolic fluxes, which is a measure of metabolite flows, can be estimated by quantitatively fitting a metabolic model to the metabolite data [46]. A simple ^13^C metabolic flux analysis model was used, comprising of glycolysis, pentose-phosphate pathway and TCA cycle [47]. The metabolite data used for the fit included the isotopomer abundances of extracellular pyruvate and lactate, and the enrichment fractions of intracellular pyruvate, lactate, malate, 2-oxoglutarate, citrate, succinate, alanine and aspartate measured by GCMS [48]. Fluxes were then estimated by least-square optimization such that simulated results gave the best fit to the experimental data. Due to the lack of absolute abundance data for intracellular metabolites, metabolite data were simulated under the assumptions of both metabolic (i.e., constant fluxes) and isotopic steady-state (i.e., maximum intracellular ^13^C enrichment) [46]. Flux changes due to Snail overexpression was quantified by Monte-Carlo analysis [46]. In this bootstrapping approach, the dataset was repeatedly corrupted with Gaussian noise 200 times, and fluxes were re-estimated each time. The resulting flux distributions were then used to quantify flux changes. Full ^13^C flux analysis results are provided within the Appendix A.

### 2.15. Statistical Analysis

Unless indicated otherwise, results comparing vector infected and Snail over-expressing Panc1 or HPDE cells were analyzed by student t-test and expressed as means ± standard error of the mean (SEM). Statistical significance was set at *p* < 0.05.

## 3. Results

### 3.1. Comparison of Basal Levels of EMT Markers in Panc1 and HPDE Cells Establishes EMT Status in Panc1 Cells

Prior to generation of Snail overexpressing Panc1 and HPDE cell lines, we first sought to determine their basal levels of EMT status. To achieve this, we performed immunoblotting on both Panc1 and HDPE cells cultured under normal conditions to look at basal markers of EMT status, including E-cadherin, N-cadherin, and vimentin (Figure 1). These preliminary immunoblotting experiments confirmed that Panc1 cells are natively somewhere along the EMT spectrum, displaying both markers of epithelial cell type (E-cadherin) as well as markers of mesenchymal status. Conversely, HPDE cells only displayed markers of epithelial status, indicating little to no induction of EMT.

### 3.2. Snail Overexpression Induced EMT in Panc1 and HPDE Cells

To study the metabolic changes associated pancreatic cells either already on the EMT spectrum or pancreatic cells with little EMT induction, we overexpressed the principal EMT-inducing transcription factor Snail in the PDAC cell line Panc1 and in non-tumorigenic HPDE cells respectively. Cells were infected with either the empty retroviral pBabe-puro vector (vector) or vector containing human SNAI1 (Snail). Two weeks after puromycin selection, surviving cells of the Snail clones in both cell lines displayed distinct morphology compared to the vector control in that they were more spindle like and dispersed, suggesting the dissociation of tight junctions (Figure 2A or Figure 2E). In Panc1, the increase in Snail (15-fold, *p* < 0.01) was coupled with marked reductions of E-cadherin levels (*p* < 0.001) in Snail-overexpressed cells, while levels of mesenchymal markers (N-cadherin and vimentin) presented little change (Figure 2B). In HPDE cells, N-cadherin and vimentin, as well as Snail, were only present at negligible levels in vector control but were remarkably induced upon Snail overexpression (80-fold increase, Figure 2F). The overexpression of Snail in HPDE also resulted in significant decreases in E-cadherin levels (Figure 2F).

To assess the functional effect of Snail overexpression in terms of migratory capacity, vector control and Snail-overexpressed cells were subjected to wound healing assays. Migration as indicated by the area of wound closure over 24 hours surprisingly did not differ between vector and Snail Panc1 cells (Figure 2C), while Snail resulted in increased percentage of wound closure over 24 hours in HPDE cells (*p* < 0.05, Figure 2G). The proliferation of cells over a 4-day period, measured by crystal violet assays, was slightly but significantly (*p* < 0.01) slowed down by Snail overexpression in Panc1 but not in HPDE cells (Figure 2D or Figure 2H).

### 3.3. Snail Overexpression Resulted in Increased Glucose Uptake and Lactate Secretion in Panc1 Cells

The upregulation of aerobic glycolysis, or increased glycolysis and lactate production in the presence of sufficient oxygen, has been frequently observed during tumorigenesis and in some cancer cells undergoing EMT [26,27,32,34,35,36]. It was also one of the most pronounced changes seen with TGFβ-induced EMT in Panc1 cells, during which Snail was induced [41]. Here the overexpression of Snail in Panc1 cells also resulted in the upregulation of glucose uptake by nearly 2-fold (*p* < 0.01, Figure 3A). There were no associated changes in SLC2A1 (encoding Glut1), SLC2A3 (encoding Glut3) or HK2, the enzyme converting glucose to glucose-6-phosphate (Figure 3D–F). Snail overexpression in Panc1 cells also caused a 2-fold increase in secreted lactate over a 3-day period (*p* < 0.01, Figure 3B). Using the Seahorse XF96 Analyzer system, lactate production was measured in a more acute setting as the basal extracellular acidification rate sampled over 20 min (Figure 3C). The basal extracellular acidification rate (ECAR; i.e., first 20 mins of assay) was slightly higher in Snail overexpressing Panc1 cells (*p* < 0.05, Figure 3C,D), while no difference was reported in maximal glycolysis after inhibition of mitochondrial ATP production by oligomycin. This lack of difference maximal glycolysis meant that Snail-overexpressing Panc1 cells had lower glycolytic capacity (i.e., the difference between maximal and basal glycolysis; *p* < 0.05; Figure 3C,D). Despite the changes in lactate production, no alterations were seen in LDH-A levels or LDH-B, MCT1 expressions whereas MCT4 expression was enhanced slightly (*p* < 0.05) (Figure 3F–I). 

### 3.4. Snail Overexpression Resulted in Increased Glucose Uptake and Lactate Production in HPDE Cells

HPDE cells overexpressing Snail displayed elevated glucose uptake to a level comparable to Panc1 cells (2-fold, *p* < 0.001, Figure 4A). The augmentation of glucose uptake was accompanied by reduced SLC2A1 (Glut1) expression (*p* < 0.001, Figure 4D) and a 4-fold increase in SLC2A3 (Glut3) expression (*p* < 0.01, Figure 4E). The level of HK2 did not differ between vector and Snail clones (Figure 4F). Although levels of lactate production in vector control HPDE cells were higher than that observed for vector control Panc1 cells, Snail-overexpressing HPDE cells also showed an increase in lactate accumulation in the culture media (Figure 4B). This increase in lactate output was also apparent over the 30-minute period in which basal ECAR was measured using the Seahorse XF96 Analyzer (Figure 4C,D). Similar to Snail overexpressing Panc1 cells, there was no difference observed in maximal glycolysis rate leading to an overall decrease in glycolytic capacity in Snail overexpressing HPDE cells (*p* < 0.05). Among the lactate production (LDH-A and B) and secretion (MCT1 and 4) mRNA measured, only LDH-B transcript displayed a nearly 2-fold increase (*p* < 0.001) (Figure 4F–I). 

### 3.5. Snail Overexpression Impacted on Oxidative Metabolism in Both Panc1 and HPDE Cells

Following the observations of enhanced aerobic glycolysis, we next investigated the overall and glucose-specific oxidative metabolism in Panc1 and HPDE cells. Using the Seahorse XF96 Analyzer, OCR at basal levels were not different in either the Panc1 or HPDE Snail overexpressing cells compared with their respective vector controls. Maximal OCR (after addition of the mitochondrial uncoupler FCCP that elicits maximal respiration) was significantly reduced upon Snail-overexpression-induced EMT in both Panc1 (*p* < 0.001) and HPDE (*p* < 0.05) cells (Figure 5B or Figure 5E). Notably, the maximally stimulated OCR in Snail-overexpressed Panc1 cells was nearly halved in comparison to vector control and was not higher than its basal level (Figure 5A). Alongside decreases in maximal respiration in Snail overexpressing Panc1 cells (*p* < 0.001), significant decreases were also observed in ATP production (*p* < 0.05), spare respiratory capacity (*p* < 0.001), and non-mitochondrial respiration (*p* < 0.01, Figure 5B). These observations were in line with significant (*p* < 0.05) decreases in the content of mitochondrial ETC subunits V, III, II and I in Snail-overexpressing Panc1 cells (Figure 5C). Glucose-specific oxidation measured using ^14^C-labelled glucose tracers was, however, not altered in Panc1 (Figure 5D). 

In contrast to Panc1, the OCR at basal level and after addition of oligomycin, FCCP and rotenone remained unaffected by Snail overexpression in HPDE cells (Figure 5E). Snail overexpressing HPDE cells displayed a lower maximal respiration compared with vector control (*p* < 0.05), and a trend towards decreased spare respiratory capacity (Figure 5F). The decrease observed in maximal respiration observation was in line with reductions in the levels of ETC complexes in HPDE cells upon Snail-driven EMT (*p* = 0.088 for complex III; *p* < 0.01 for complex I), albeit to a lesser extent than Panc1 (Figure 5G). Glucose specific oxidative activity was unaltered in Snail overexpressed cells, as indicated by the ^14^CO_2_ produced from ^14^C-glucose substrate (Figure 5H). 

### 3.6. ^13^C Flux Analysis Validated Observed Changes in Aerobic Glycolysis and TCA Cycle Activity

A modelling approach was used to provide a coherent interpretation of metabolic fluxes using metabolite data obtained for Panc1 and HPDE cells with or without Snail overexpression (Figure 6A). Fluxes were estimated by fitting the metabolic model to the measured accumulation rate of extracellular lactate and pyruvate, and to the ^13^C enrichment pattern of intracellular metabolites (Figure 6B,C). Despite similarities in the enrichment fractions (Figure 6B), our analyses accounted for the fact that media used between the two cell lines (i.e., DMEM and KSF media) were different. In DMEM, glucose was 95% labelled and lactate was present, whereas in KSF media glucose was only 59% labelled and had no lactate (Figure 6A or Figure 6C). Similarly, only DMEM contained free alanine (results not shown), which explained the significant dilution of ^13^C-enriched intracellular alanine in Panc1 compared to HPDE (Figure 6B). The assimilation of unlabeled pyruvate was added ad-hoc to better fit metabolite data from HPDE cells.

Flux results confirmed that aerobic glycolysis increased in Snail overexpression, for both Panc1 and HPDE cells (Figure 6D). Basal lactate secretion rates were lower in Panc1 than HPDE cells; this is consistent with the results from the enzymatic lactate assay and ECAR (Figure 3B–D and Figure 4B–D). Likewise, the reduction of basal OCR in Panc1 but not HPDE (Figure 5A,E) was reproduced by the estimated pyruvate dehydrogenase fluxes (Figure 6D). Only Panc1 cells with Snail overexpression showed a net consumption of extracellular pyruvate (Figure 6C), although all cultures were secreting pyruvate from glucose.

Flux modelling revealed a few features not immediately observed from metabolite data. The overexpression of Snail in Panc1 cells reduced TCA cycle activity to a greater extent than HPDE cells. This was achieved by lowering both pyruvate dehydrogenase flux and complete oxidation of glutamine, with the latter indicated by a reduced mitochondrial malic enzyme activity and a concomitant shift towards cytoplasmic malic enzyme flux (Figure 6D). This metabolic configuration was required to reproduce the reduced enrichment of intracellular pyruvate and lactate, which could not be solely accomplished by the reversible exchange between the respective intracellular and extracellular pools. Overall, ^13^C flux analysis showed that Snail overexpression increased aerobic glycolysis and altered carbon flow in the TCA cycle, more so in Panc1 cells than in HPDE.

### 3.7. Snail Overexpression in Panc1 Cells Does Not Increase Resistance to Gemcitabine or Combination Gemcitabine-Paclitaxel Therapy 

Given the known role of EMT in chemoresistance in PDAC [10], we sought to determine if Snail-overexpression and subsequent EMT induction in Panc1 cells could increase resistance to chemotherapies commonly used in PDAC treatment. To this end, we determined the half-maximal inhibitory concentration (IC_50_) of gemcitabine, paclitaxel and gemcitabine combined with an IC_75_ dose of paclitaxel (Table 1). Overexpression of Snail in Panc1 cells did not result in an increased resistance to gemcitabine alone or in combination with an IC_75_ dose of paclitaxel, but Snail-overexpressing Panc1 cells were slightly more sensitive to paclitaxel monotherapy. We hypothesized that Snail overexpression may give PDAC cells enhanced chemoresistance under the low nutrient microenvironment conditions typically found in PDAC tumors as a result of the dense stromal/desmoplastic reaction [49], but no statistically significance differences were observed between the IC_50_ of Snail-overexpressing and vector controls treated with combination therapy under limiting glucose conditions.

## 4. Discussion

The occurrence of EMT in response to micro-environmental factors partially underlines the malignant phenotype and chemoresistance of PDAC. High levels of Snail, a potent EMT-inducing transcription factor, closely correlate with lymph node invasion and distant metastasis in human PDAC samples [10,11,12,50,51,52]. Attenuation of Snail expression in PDAC cell lines resulted in the reversal of EMT, together with decreased sphere and colony formation capacity [16]. It has been shown in several studies that PDAC cells overexpressing Snail underwent EMT and exhibited EMT-associated invasive behaviors both in vitro and in vivo [10,13,14,15]. In the present study, stable overexpression of Snail in the PDAC cell line Panc1 and non-tumorigenic HPDE cells resulted in pronounced EMT-like phenotypic change as evidenced by alterations in morphology, epithelial/mesenchymal markers and, in the case of HPDE, enhanced migratory capacity (Figure 2). There were also several adaptations in glucose and oxidative metabolism observed with snail overexpression in both cell lines. Despite these alterations to metabolism, Snail-overexpression in PDAC cells did not result in enhanced resistance to gemcitabine or combination gemcitabine/paclitaxel therapy when cultured under either high or limited glucose conditions (Table 1). 

Over the past decade, metabolic reprogramming has been increasingly recognized as a hallmark of oncogenic transformation in PDAC and other cancers [17,18,19]. A small body of literature has also emerged in the last 5 years uncovering additional metabolic alterations related to EMT in breast, lung, ovarian, cervical and prostate cancers, but the actual changes vary considerably across different EMT models and cancer types [26,27,28,29,30,31,32,33,34,35,36,37,38,39,40]. The EMT events were mostly accompanied by elevated levels of more than one EMT-inducing transcription factors, with Snail being a principal player in the majority of cases [28,29,30,31,32,33,36]. In the context of PDAC, Snail was highly induced in Panc1 during TGFβ induced EMT, which was associated with upregulated aerobic glycolysis [41]. The induction of EMT by Snail overexpression in the current study was also accompanied by augmented glucose uptake, lactate production and increased levels of basal glycolytic activity, with changes in the expression of transporters and enzymes involved in these processes, namely increased MCT4 expression in Panc1 and higher GLUT3 as well as LDH-B expression in HPDE cells (Figure 3 and Figure 4). In addition, marked downregulations of mitochondrial ETC subunits content and, particularly in the case of Panc1, impaired overall oxidative metabolism were evident.

A direct role of Snail in regulating the glycolytic process has been reported in several occasions. Dong et al. (2013) observed an inverse correlation between levels of Snail and the gluconeogenic enzyme FBP1 in breast cancers [27], hence favoring glucose flux through glycolysis rather than the reverse direction. In basal-like breast cancer, an aggressive subtype containing abundant EMT features, the Snail-G9a-Mnmt1 complex was shown to directly bind to the FBP1 promoter, leading to DNA methylation and transcriptional silencing of the gene [27]. The Snail-mediated suppression of FBP1 was thought to promote glucose uptake and lactate production via improving insulin sensitivity and decreasing PDH (the enzyme catalyzing the conversion of pyruvate to acetyl-CoA and therefore mitochondrial oxidation) activity [27]. In the same vein, Snail levels were high in the more aggressive and castration-resistant subtype of prostate cancer, in which Snail depletion reduced glucose consumption and lactate production [38]. Snail was found to regulate metabolism through miRNA-126-mediated RPS6KB1/HIF1α/PKM2 signaling [38]. In Madin Darby Canine Kidney (MDCK) cells, Snail overexpression resulted in increased activity of phosphofructokinase, a rate-limiting glycolytic enzyme promoting the opposite process to FBP1 [37]. Increased PDK1 expression and the consequent reduction of PDH activity also pointed to the diversion of glycolytic flux towards lactate synthesis [37]. The glycolytic switch was also observed in the breast cancer cell lines MCF-7 and MDA-MB-231 when Snail was induced by Wnt signaling or E-cadherin knock-down [39]. 

The augmentations of glucose uptake, lactate production, and basal glycolytic activity seen in both Panc1 and HPDE, despite differences in molecular changes, could contribute to EMT-related functional properties especially increased migratory and invasive potential. Enhanced aerobic glycolysis has been associated with invasive cancers and several glycolytic enzymes have been shown to stimulate migration via signaling effects [53,54,55]. The strongest argument in favor of the glycolytic dependency of migration came from observations that mesenchymal prostate and breast cancer cells exhibited higher aerobic glycolysis, cytoskeletal remodeling and faster migration than epithelial counterparts while no difference in mitochondrial ATP production was found [56]. Migration was attenuated only by inhibition of glycolysis but not mitochondrial respiration [56]. As cell migration is an energy-expensive process involving major remodeling of the cytoskeletal network [57,58], one benefit of the profound upregulation of the glycolytic pathway in PDAC EMT models is presumably to maintain a steady and rapid supply of ATP for cellular migration. The increased lactate secretion by tumor cells could result in an acidic peri-tumor microenvironment which induces MMP-9 expression and the release of other proteolytic enzymes to degrade components of the ECM [59,60]. This is of particular importance in PDAC, which exhibits a prominent desmoplastic reaction involving extensive proliferation of stromal cells and ECM deposition, constituting a physical barrier for tumor cell extravasation [49]. The flow of H^+^ along its concentration gradient to adjacent normal tissues could also lead to toxic effects in normal cells such as stromal cells but not cancer cells that developed resistance to low pH environment during carcinogenesis [61]. In addition, lactate has been reported to directly enhance tumor cell motility and contribute to tumor immune escape by inhibiting monocyte migration and cytokine release [62]. However, overexpression of Snail within a metastatic subclone of PC-3 prostate cancer cells reduced both glucose and lactate consumption and increased oxidative metabolism, indicating that expression of EMT features may not always coincide with a higher glycolytic phenotype across all cancer types [63].

There was evidence of downregulation of ETC complex subunits in both cell lines with Snail overexpression. These changes were more pronounced in Panc1 cells compared to HPDE and translated into a functional deficit where both the maximal OCR and the flow of carbons into the TCA cycle decreased in Panc1 cells. The inhibition of oxidative phosphorylation by Snail was implicated in Dong et al. (2013) where Snail-mediated reduction in FBP1 resulted in the loss of mitochondrial transcription factor B1M (TFB1M), leading to defects of protein translation in ETC complex I components [27]. As complex I and III are the main sites of ROS production, the downregulation of complex I level caused by Snail was accompanied by ROS reduction [27]. Lee et al. (2012) showed the direct binding of Snail to promoters of three Cytochrome c oxidase (COX) subunits of ETC complex IV [39]. Complex IV activity and mitochondrial respiration were impaired as a result but no change in ATP status was seen, possibly owing to the compensatory increase in glycolytic ATP production [39]. Given that cell migration is an energy consuming process and the increased wound closure was only observed in HPDE cells overexpressing Snail that exhibit only a small change in maximal respiration (Figure 5F), one could speculate that the large decrease in maximal respiration and ATP production seen in Panc1-Snail cells limited their migratory capacity. This phenomenon was observed in other models of EMT induction where loss of oxidative metabolism impedes cell migration [64]. The dissociation of mitochondrial ETC content and basal/maximal OCR in HPDE cells with or without Snail might be attributed to decreased electron donation to the ETC complexes and increased energy consumption in other EMT-related processes. Alternatively, that fact that Panc1 cells are already on the EMT spectrum prior to Snail induction may limit any additional gain in migratory capacity (Figure 1).

There have been indications in the literature that the process of EMT promotes chemoresistance in various carcinomas including PDAC [65,66,67,68]. While the induction of several transcription factors and increased stemness were suggested as possible mechanisms, it is not clear if EMT-associated metabolic programming or increased glycolysis plays a role [66,67]. In the present study, Snail-induced EMT changes did not alter Panc1 sensitivity to gemcitabine or combination gemcitabine paclitaxel treatment despite marked enhancement of glycolysis and slight reduction on proliferation (Table 1 and Figure 1). Further work is required to probe the effects of EMT-related metabolic reprogramming on chemo-sensitivity in additional PDAC cell lines and with other first-line chemotherapeutic agents.

## 5. Conclusions

Collectively, Snail overexpression in the PDAC cell line Panc1 and in non-tumorigenic HPDE cells resulted in the induction of EMT and a range of accompanying metabolic changes. In both cell lines Snail overexpression resulted in increased glucose uptake and lactate production, as well as reductions in mitochondrial ETC protein content. Additionally, Snail overexpression caused decreased carbon flux from glucose in the TCA cycle in Panc1 cells only, with no change in Snail-overexpressing HPDE cells. Despite the induction of EMT status and detection of metabolic reprogramming of glucose metabolism in Panc1 cells Snail-overexpression did not result in enhanced resistance to gemcitabine or combination gemcitabine/paclitaxel therapy. This work highlights the role that Snail plays as an effector of EMT and its role in the induction of metabolic reprogramming in PDAC. Further research to uncover specific changes in metabolic enzymes, pathways and energetic profiles that are essential to EMT in PDAC is required to allow therapeutic interventions from a metabolic angle.

## Figures and Tables

**Figure 1 jcm-08-00822-f001:**
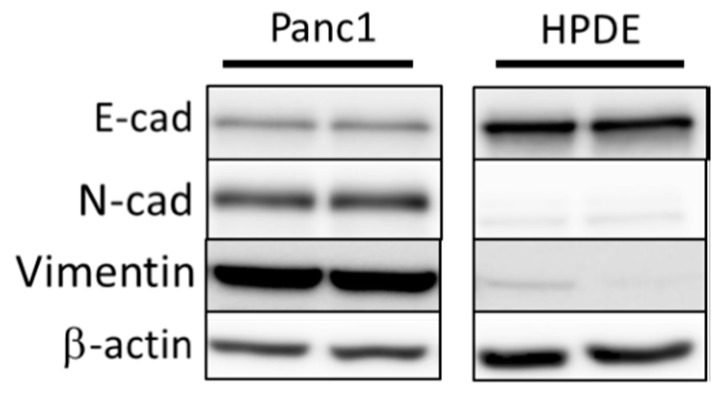
Immunoblotting of basal levels of EMT markers E-cadherin (E-cad), N-cadherin (N-cad), and vimentin in Panc1 and HPDE cells. β-actin was used as loading control.

**Figure 2 jcm-08-00822-f002:**
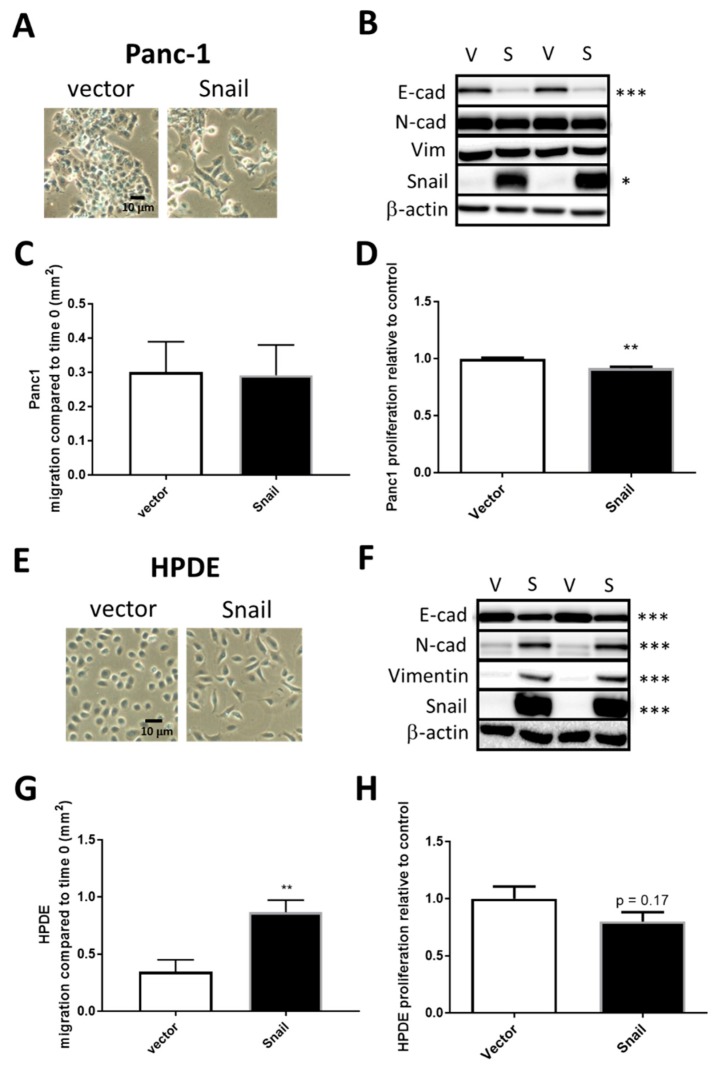
Snail overexpression induced EMT in Panc1 (**A**–**D**) and HPDE (**E**–**H**) cells. Vector control (V) and Snail-overexpressing (S) cells were generated in Panc1 via retroviral-mediated infections. (**A**,**E**) Representative cell images were taken under bright field microscopy. (**B**,**F**) Cell lysates were resolved by SDS-PAGE and immunoblotted with anti-E-cad, anti-N-cad, anti-vimentin, and anti-Snail antibodies with β-actin used as a loading control. (**C**,**G**) Cell migration as measured by wound healing assay. (**D**,**H**) Cell proliferation as measured by crystal violet assay. Results are shown as mean ± SEM with *n* = 3. * *p* < 0.05, ** *p* < 0.01, *** *p* < 0.001 for difference between vector control and Snail-overexpressing cells.

**Figure 3 jcm-08-00822-f003:**
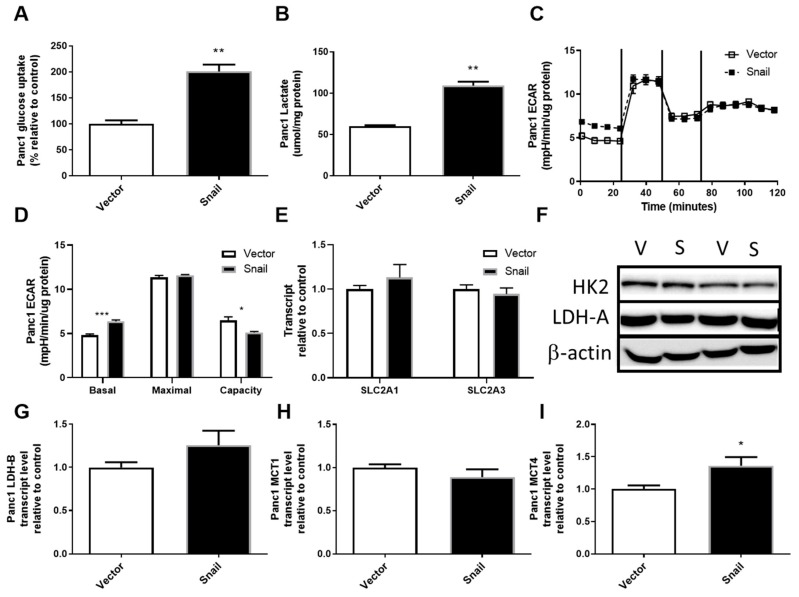
Effects of Snail overexpression on glucose uptake and lactate production in Panc1 cells. (**A**) Rate of glucose uptake measured using by ^3^H-2-Deoxy-Glucose uptake over an 8 min period. (**B**) Lactate assay performed on cell culture media after 72 hours of culture. (**C**) Extracellular acidification rate (ECAR) measured using the Seahorse XF96 Analyzer at basal levels for 30 min followed by sequential injections of 1 μM oligomycin, 0.5 μM Carbonyl cyanide-4-(trifluoromethoxy)-phenylhydrazone (FCCP) and 1 μM rotenone at 30 min intervals. (**D**) basal and maximal glycolytic activity and glycolytic capacity calculated from Seahorse data, (**E**) Fold-change in total RNA for SLC2A1 (encoding Glut1) and SLC2A3 (encoding Glut3) measured by qPCR using β-actin as housekeeper, (**F**) Immunoblotting results for hexokinase II (HK2) and lactate dehydrogenase A (LDH-A) with β-actin as loading control. (**G**–**I**) Fold change in total RNA detected for lactate dehydrogenase-B (LDH-B) and monocarboxylate transporter 1 (MCT1) and 4 (MCT4). Results are shown as mean ± SEM with *n* = 3. * *p* < 0.05, ** *p* < 0.01 for difference between vector control and Snail-overexpressing cells.

**Figure 4 jcm-08-00822-f004:**
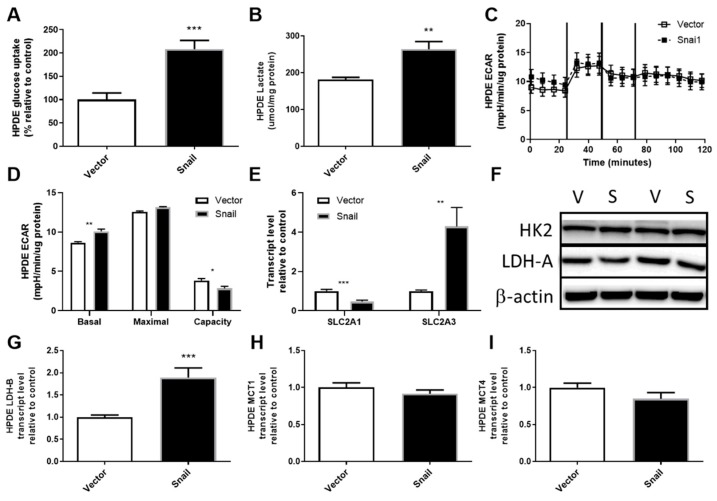
Effects of Snail overexpression on glucose uptake and lactate production in HPDE cells. (**A**) Rate of glucose uptake measured by ^3^H-2-Deoxy-Glucose tracer over an 8 min period. (**B**) Lactate assay performed on cell culture media collected after 72 hours of culturing. (**C**) Extracellular acidification rate (ECAR) measured using the Seahorse XF96 Analyzer at basal levels for 30 min followed by sequential injections of 1 μM oligomycin, 0.5 μM Carbonyl cyanide-4-(trifluoromethoxy)-phenylhydrazone (FCCP) and 1 μM rotenone at 30 min intervals. (**D**) basal and maximal glycolytic activity and glycolytic capacity calculated from Seahorse data. (**E**) Fold-change in total RNA for SLC2A1 (encoding Glut1) and SLC2A3 (encoding Glut3) measured by qPCR using β-actin as housekeeper. (**F**) Immunoblotting results for hexokinase II (HK2) and lactate dehydrogenase A (LDH-A) with β-actin as loading control. (**G**–**I**) Fold change in total RNA detected for lactate dehydrogenase-B (LDH-B) and monocarboxylate transporter 1 (MCT1) and 4 (MCT4). Results are shown as mean ± SEM with *n* = 3. ** *p* < 0.01, *** *p* < 0.001 for difference between vector control and Snail-overexpressing cells.

**Figure 5 jcm-08-00822-f005:**
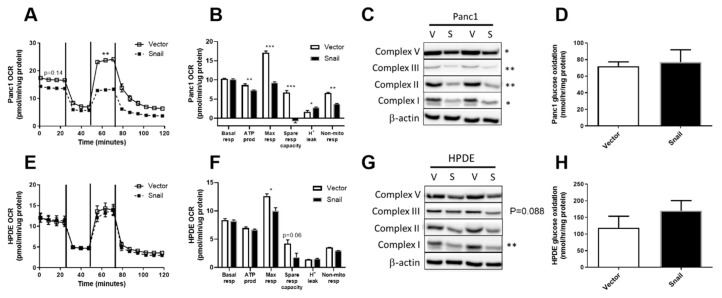
Effects of Snail overexpression on oxidative metabolism in Panc1 and HPDE cells. (**A**,**E**) Oxygen consumption rate (OCR) was measured using the Seahorse XF96 Analyzer at basal levels for 30 min followed by sequential injections of 1 μM oligomycin, 0.5 μM Carbonyl cyanide-4-(trifluoromethoxy)-phenylhydrazone (FCCP) and 1 μM rotenone at 30 min intervals. Basal and maximal (following FCCP injection) values of OCR from the time-course data were used for statistical analysis. (**B**,**F**) Basal respiration, ATP production, maximal respiration, spare respiratory capacity, proton leak and non-mitochondrial respiration calculated from Seahorse trace data for Panc1 and HPDE cells respectively. (**C**,**G**) Immunoblotting results for ETC complex I, II, III, V antibodies, with β-actin used as loading control. Densitometry on western blots was performed using image J. (**D**,**H**) Glucose oxidation was measured using the U-^14^C-Glucose tracer over one-hour period. Results are shown as mean ± SEM with *n* = 3. * *p* < 0.05, ** *p* < 0.01 for difference between vector control and Snail-overexpressing cells.

**Figure 6 jcm-08-00822-f006:**
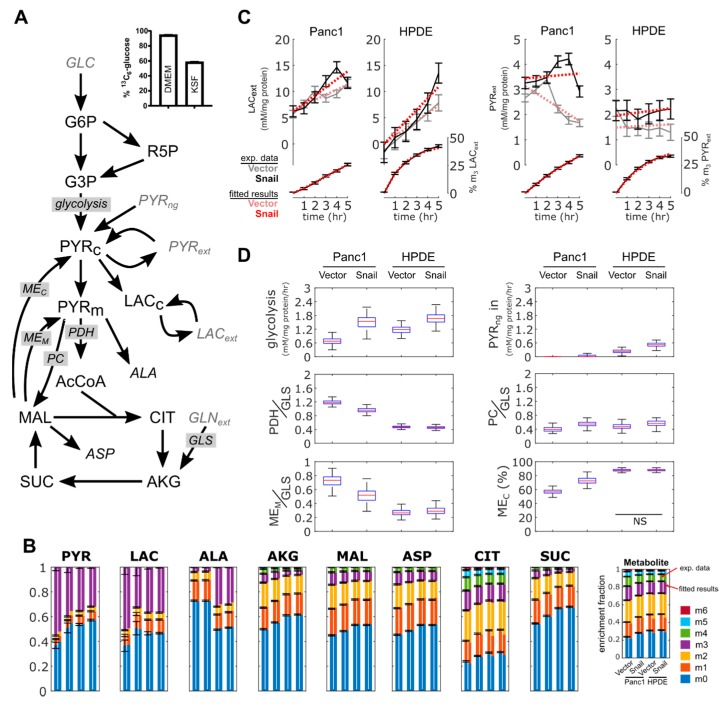
Metabolomics and flux analysis of the effects of Snail overexpression on glucose metabolism in Panc1 and HPDE cells. (**A**) 13C metabolic flux analysis model used to interpret metabolite data. Input substrates shown as gray. Average glucose ^13^C-enrichment in DMEM and KSF media shown to highlight differences in starting label. (**B**) ^13^C-enrichment of intracellular metabolites, showing measured fractional enrichments (left bars) and simulated results from the best fit (right bars). Data are shown with mean ± 1 SD with *n* = 3. (**C**) Abundance and ^13^C-enrichment trajectories of extracellular lactate and pyruvate. “m3” represents fraction of lactate and pyruvate labelled at all three carbons. Results from the best fit are shown as red dotted lines. Error bars represent ± 1 SD. (**D**) Box-and-whiskers plots show estimated fluxes. PDH, PC and ME_M_ fluxes normalized to glutamine uptake flux. ME_C_ flux expressed as a fraction of total ME flux. All flux changes were significant, unless indicated otherwise. Enzyme: ME_C_ (cytoplasmic malic enzyme), ME_M_ (mitochondrial malic enzyme), PC (pyruvate carboxylase), GLS (glutaminase), glycolysis (pyruvate kinase), PDH (pyruvate dehydrogenase). Metabolites: PYR (pyruvate), LAC (lactate), ALA (alanine), AKG (2-oxoglutarate), MAL (malate), ASP (aspartate), CIT (citrate), SUC (succinate), R5P (ribose 5 phosphate), AcCoA (acetyl-CoA), GLN (glutamine). NS *p* > 0.05.

**Table 1 jcm-08-00822-t001:** The half-maximal inhibitory concentration (IC_50_) of gemcitabine, paclitaxel and combination gemcitabine with an IC_75_ dose of paclitaxel in vector control and Snail overexpressing Panc1 cells. Combination treatment was given under both high and no glucose media conditions.

Panc1	Gemcitabine	Paclitaxel *	Gemcitabine + IC _75_ Paclitaxel	
			High Glucose	No Glucose
Vector	1.8 (1.2–2.9) × 10^−7^	3.3 (2.4–4.6) × 10^−9^	1.3 (0.5–4.2) × 10^−7^	1.3 (0.3–5.3) × 10^−7^
Snail	1.4 (0.8–2.9) × 10^−7^	2.0 (1.3–3.1) × 10^−9^	3.0 (1.1–8.6) × 10^−7^	6.3 (0.4–52.2) × 10^−7^

IC_50_ (M) with 95% confidence interval. Vector compared with Snail overexpression: * *p* < 0.05.

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
