# Peer review of "Snail-Overexpression Induces Epithelial-mesenchymal Transition and Metabolic Reprogramming in Human Pancreatic Ductal Adenocarcinoma and Non-tumorigenic Ductal Cells"

_jcm, 2019, doi:10.3390/jcm8060822_

Round 1
Reviewer 1 Report
The manuscript addresses an important issue, namely Snail induced EMT and metabolic reprogramming in pancreatic cancer. While the EMT data are not novel/convincing or associated in anyway with the findings related to metabolic reprogramming there are some potentially interesting observations findings regarding the latter. Overall, the study at its present form seems premature.
The rationale for choosing this specific PDAC line (PANC1) versus this specific non-tumorigenic immortalized pancreatic ductal epithelial cell line (HPDE) is not clear
There are no data regarding levels of Snail expression in control and overexpressing lines. How do these compare ?
The evidence that Snail overexpression results in EMT is convincing in the case of HPDE cells but not for PANC1 cells. In the latter every single marker or assay they used, the only expression being that of E-cad, does not support the induction of EMT. It is possible that PANC1 cells are refractory to EMT induction by Snail. In any case, Snail is widely associated with EMT, these data here seem superfluous. Additionally, the attempted association of EMT with metabolic reprogramming subsequently, is very weak. If anything, form these data it can be argued that Snail mediated metabolic reprogramming can happen independently of EMT !
The data in figures 2 and 3 are presented in a confusing way. The authors should make an attempt to integrate these data, particularly with the aim to compare the basal (vector) state of PANC1 and HPDE as well as the Snail induced levels. The Seahorse data would benefit for the additional calculation and graphical presentation of basal glycolytic activity as well as glycolytic capacity and how these change upon Snail expression. Transcriptional changes are much more pronounced in the HPDE cells but the change in basal glycolytic activity appears very similar. The authors make no attempt to explain this.
It would be more informative if the authors also present basal respiration as well as respiratory capacity w/ and w/o Snail overexpression using an additional graph (Figure 4). Again here, there is the paradox that while levels of respiratory complexes are reduced in both PANC1 and HPDE cells upon Snail overexpression. The authors make no attempt to explain this.
It is unclear how the ‘Snail effects on lipid metabolism’ tie with all the other data in the paper.
The flux analysis is not well explained. What is the ‘modelling approach’ used (figure 6C) ? Importantly, the changes in metabolic flux presented in 6B, D, are they statistically significant ?
It would be helpful if paragraph titles present a conclusion rather than a non-committal statement.
Cat no for the antibodies used should be provided.
Author Response
Reviewer 1
The manuscript addresses an important issue, namely Snail induced EMT and metabolic reprogramming in pancreatic cancer. While the EMT data are not novel/convincing or associated in anyway with the findings related to metabolic reprogramming there are some potentially interesting observations findings regarding the latter. Overall, the study at its present form seems premature.
The rationale for choosing this specific PDAC line (PANC1) versus this specific non-tumorigenic immortalized pancreatic ductal epithelial cell line (HPDE) is not clear
We have amended the introduction of the manuscript to make it clearer as to why we have chosen to study Snail overexpression specifically in Panc1 and HPDE cells. Our reasons for studying the effects of Snail overexpression in these cell lines stem from past work by our laboratory showing that treatment of Panc1 cells with either TGFβ or TGFβ+TNFα induced EMT alongside concurrent induction of Snail (Liu et al. 2016 Cancer & Metabolism). Therefore, in the present work we chose to study the specific effects of Snail overexpression within both Panc1 cells, which contain markers both epithelial and mesenchymal status (indicating that they are partially differentiated along the EMT spectrum), and a purely epithelial nontumorigenic pancreatic cell line, HPDE cells. We have also amended the results section to include data showing relative basal levels expression of markers of both epithelial and mesenchymal and phenotypes in non-transfected Panc1 and HPDE cells to further support the premise of this work.
There are no data regarding levels of Snail expression in control and overexpressing lines. How do these compare ?
We have added in-text the fold change measured by densitometry in Snail expression in overexpression vs. vector control for both cell lines.
The evidence that Snail overexpression results in EMT is convincing in the case of HPDE cells but not for PANC1 cells. In the latter every single marker or assay they used, the only expression being that of E-cad, does not support the induction of EMT. It is possible that PANC1 cells are refractory to EMT induction by Snail. In any case, Snail is widely associated with EMT, these data here seem superfluous. Additionally, the attempted association of EMT with metabolic reprogramming subsequently, is very weak. If anything, form these data it can be argued that Snail mediated metabolic reprogramming can happen independently of EMT !
The lack of induction of EMT characteristics such as increased migratory capacity and cell proliferation in Panc1 cells can be readily explained. As described above, Panc1 cells display markers of both epithelial and mesenchymal cells, indicating that they are more progressed on the EMT spectrum than the non-tumorigenic HPDE cells. Indeed, the small amount of evidence available suggests that Panc1 cells have enhanced inherent migratory capacity compared with other PDAC cells lines (Deer et al. 2010 Pancreas), and it is therefore likely that Snail-overexpression is unable to markedly enhance this further. Despite the lack of changes to either cell proliferation or migration, we saw a significant decrease in the expression of E-cadherin in Snail overexpressing Panc1 cells (Figure 2; formerly Figure 1), indicating further mesenchymal differentiation. Alongside this, we saw a clear shift in metabolic phenotype within Panc1 cells with Snail overexpression that is consistent with previously published work on inducing EMT both from our laboratory (Liu et al. 2016 Cancer Metab) and others (Dong et al. 2013 Cancer Cell; Kim et al. 2017 Nature Comm). Therefore, the data included herein is definitely in line with previously reported shifts in glycolytic activity with Snail-overexpression and EMT status, and provides further support to the idea that Snail overexpression can drive adaptive metabolic reprogramming within cancer cells.
The data in figures 2 and 3 are presented in a confusing way. The authors should make an attempt to integrate these data, particularly with the aim to compare the basal (vector) state of PANC1 and HPDE as well as the Snail induced levels. The Seahorse data would benefit for the additional calculation and graphical presentation of basal glycolytic activity as well as glycolytic capacity and how these change upon Snail expression. Transcriptional changes are much more pronounced in the HPDE cells but the change in basal glycolytic activity appears very similar. The authors make no attempt to explain this.
Figures 3 and 4 (formerly Figures 2 and 3) have been edited to make them clearer, including adjustments to y-axes to remove overlapping labels. We have also edited the figure legend to remove extraneous information. As the PANC1 and HPDE experiments were not performed in such a way that makes direct comparison accurate, we have deliberately arranged panels in Figure 2, 3 and 4 such that Panc1 and HPDE results were segregated. Both the Panc1 and HPDE cells were cultured at different times and many of the analyses were performed separately, which precludes direct comparison between the two cell lines. We have, however, at the suggestion of reviewer 2 included a new section within the results describing basal levels of EMT markers between naïve Panc1 and HPDE cells cultured and analysed within the same set of experiments. (Figure 1).
Basal and maximal glycolytic activity and glycolytic activity has been calculated for both Snail overexpressing and vector controls in both cell lines and is now presented in Figures 3 & 4 for Panc1 and HPDE cells respectively.
It would be more informative if the authors also present basal respiration as well as respiratory capacity w/ and w/o Snail overexpression using an additional graph (Figure 4). Again here, there is the paradox that while levels of respiratory complexes are reduced in both PANC1 and HPDE cells upon Snail overexpression. The authors make no attempt to explain this.
All parameters of mitochondrial function calculated from data acquired on the Seahorse have now been added to Figure 5 (formerly Figure 4). This includes basal respiration, ATP production, maximal respiration, spare respiratory capacity, proton leak, and non-mitochondrial respiration. We have likewise amended the discussion to include these new data.
It is unclear how the ‘Snail effects on lipid metabolism’ tie with all the other data in the paper.
As indicated within the manuscript, changes to lipid metabolism are an important aspect in cancer cell metabolism and there is emerging evidence that altered lipid metabolic profiles occur during EMT (Giudetti et al. 2019 Biochim Biophys Acta Mol Cell Biol Lipids, 1864:344-357). We have amended the discussion of the manuscript to include this recently published work and discuss its finding in context with that of the present study.
The flux analysis is not well explained. What is the ‘modelling approach’ used (figure 6C) ? Importantly, the changes in metabolic flux presented in 6B, D, are they statistically significant ?
The flux modelling approach was further elaborated in both Method and Results section, and we have provided references that will help readers with the methodology. We explained that “Metabolic fluxes, which is a measure of metabolite flows, can be estimated by quantitatively fitting a metabolic model to the metabolite data”. We also included a brief description of the Monte-Carlo method used to generate the flux distributions shown in Figure 7D. Flux changes that were statistically significant are now shown. We have indicated in the figure and figure legend that all flux changes were significant, except for the contribution of cytoplasmic malic enzyme. Panel B is to show the enrichment pattern of metabolites and the goodness-of-fit of model results, thus do not require significance testing.
It would be helpful if paragraph titles present a conclusion rather than a non-committal statement.
We have amended relevant paragraph titles in the results section to better present the conclusions that can be drawn therein.
Cat no for the antibodies used should be provided.
We have added the catalogue numbers for all antibodies used.
Reviewer 2 Report
The manuscript titled “Snail-overexpression induces epithelial-2 mesenchymal transition and metabolic reprogramming in human pancreatic ductal adenocarcinoma and non-tumorigenic ductal cells” by Menghan Liu et al., were aimed to investigate the effect of a very well studied transcription factor “Snail” expression in epithelial to mesenchymal transition in cancerous Panc1 cells and non-cancerous (HPDE) cell line. Authors have performed and analyzed the data meticulously and the manuscript is written well. Authors should consider improving their manuscript with the following minor modifications for the readers to get clear insights.
1. The light microscopy images of cell cultures images shown in Fig 1 are not informative and also are not clear. I assume that authors might have performed these experiments in triplicates. If so, they should produce some high quality light or confocal images upon immunofluorescence studies (with EMT markers) showing the transition from epithelial to mesenchymal morphology.
2. Authors should clearly mention in the methods section about, how long the cells were cultured? what is the (%) confluence of these cell lines which were cultured in either 6 well or flasks before and after transient transfection? Also when they performed these experiments after transfection, did the authors count the # of cells in the beginning (when they seeded) and just before making cellular extracts? This is important because the # of cells play major role in the outcome of the results.
3. Authors should mention the amount of protein loaded (in the methods) and each lane in all experiments where immunoblotting was performed.
4. What is the basal level and temporal expression (both protein and mRNA) of Snail before transfection in these cell lines ? Similarly authors should check the basal level of Epithelial and mesenchymal markers that they have used in the manuscript before transfection, because, mock transfection itself can alter the gene expression in cells.
5. The protein expression of Snail is significantly high in transiently transfected cell lines. However the migration and proliferation in Panc1 cells is minimal, when compared to HPDE cells. Authors should comment on why the Panc1 cells have slow proliferation and migration compare to HDPE cells?
6. Authors should also comment on the biochemical and physiological changes what they see in in vitro studies (in both cell lines) can be comparable to in vivo conditions.
7. LDH-B protein levels should have shown for comparison with mRNA levels in both cell lines in Figures 2 and 3. Why there is more lactate accumulation in these cells, is this is derived from non-carbohydrate source (gluconeogenesis) or does it have direct correlation with increased glucose transport?
8. Increase in Steroyl co-A desaturase 1 (SCD1) levels were observed in many metabolic diseases (diabetes, atherosclerosis, obesity and hypertriglyceridemia) and tumor malignancy. Increase in the expression of steroyl co-A desaturase1 might be due to increased glucose transport into the cellular via glut receptors. Literature studies have shown that the SCD1 is located in the endoplasmic reticulum (ER). Did the authors study any ER stress related markers?
9. Authors might consider performing a comprehensive analysis such as next generation sequencing (RNAseq) and proteomic studies using in vivo (pancreatic cancer tissue from ductal origin) and in vitro studies (Panc1 cells). The data might provide an evidence of cell-type specific genes and proteins that are involved in EMT transitions.
Author Response
Reviewer 2
The manuscript titled “Snail-overexpression induces epithelial-2 mesenchymal transition and metabolic reprogramming in human pancreatic ductal adenocarcinoma and non-tumorigenic ductal cells” by Menghan Liu et al., were aimed to investigate the effect of a very well studied transcription factor “Snail” expression in epithelial to mesenchymal transition in cancerous Panc1 cells and non-cancerous (HPDE) cell line. Authors have performed and analyzed the data meticulously and the manuscript is written well. Authors should consider improving their manuscript with the following minor modifications for the readers to get clear insights.
1. The light microscopy images of cell cultures images shown in Fig 1 are not informative and also are not clear. I assume that authors might have performed these experiments in triplicates. If so, they should produce some high quality light or confocal images upon immunofluorescence studies (with EMT markers) showing the transition from epithelial to mesenchymal morphology.
Unfortunately such experiments were unable to be conducted given the short time we had to prepare our response to this review. The light microscopy pictures contained within the manuscript were obtained on the highest resolution phase-contrast light microscope available to us and represent the best images captured throughout the course of the study. Additionally, we do not believe that the inclusion of the described experimental data would change the outcomes and conclusions already present within the current manuscript.
2. Authors should clearly mention in the methods section about, how long the cells were cultured? what is the (%) confluence of these cell lines which were cultured in either 6 well or flasks before and after transient transfection? Also when they performed these experiments after transfection, did the authors count the # of cells in the beginning (when they seeded) and just before making cellular extracts? This is important because the # of cells play major role in the outcome of the results.
As described within the method section of the manuscript, the Snail overexpression we have induced within both Panc1 and HPDE cells was a stable transfection and not a transient one. Both Panc1 and HPDE cells had been passaged less than 10 times prior to stable transfection, and relative seeding densities for transfection experiments have been described within the methods section of the manuscript. All experiments conducted after Snail transfection were likewise conducted on low passage numbers to ensure fidelity of the data.
3. Authors should mention the amount of protein loaded (in the methods) and each lane in all experiments where immunoblotting was performed.
We have amended the manuscript to indicate the amount of protein loaded for all immunoblotting experiments (i.e. 20 ug total protein ).
4. What is the basal level and temporal expression (both protein and mRNA) of Snail before transfection in these cell lines ? Similarly authors should check the basal level of Epithelial and mesenchymal markers that they have used in the manuscript before transfection, because, mock transfection itself can alter the gene expression in cells.
We have amended the manuscript to include measured levels of EMT markers E-cadherin, N-cadherin and vimentin in naïve Panc1 and HPDE cells cultured under normal conditions (Figure 1). Few differences in the relative abundance of EMT markers are observed when comparing these normal cells with vector control transfected cells (Figure 2, formerly Figure 1). Unfortunately the tight turnaround required by the journal for our response limits our ability to raise cells for immunoblotting of Snail in these cell lines; however, the lack of impact of mock transfection on EMT markers within both cell lines give us confidence that Snail expression in these cells is similarly unaffected.
5. The protein expression of Snail is significantly high in transiently transfected cell lines. However the migration and proliferation in Panc1 cells is minimal, when compared to HPDE cells. Authors should comment on why the Panc1 cells have slow proliferation and migration compare to HDPE cells?
Being a pancreatic ductal adenocarcinoma cell line obtained from an excised pancreatic tumour, Panc1 cells already present on the EMT spectrum and display markers of EMT including expression of N-cadherin and vimentin prior to Snail overexpression. Therefore, Snail overexpression in Panc1 cells may not on its own impact greatly on the migration and proliferation of this cell type. We have amended the manuscript to better reflect the key differences between Panc1 and HPDE cells in terms of EMT status prior to Snail overexpression and include this hypothesis as to why no changes were seen in growth or proliferation within Snail overexpressing Panc1 cells.
6. Authors should also comment on the biochemical and physiological changes what they see in in vitro studies (in both cell lines) can be comparable to in vivo conditions.
We believe that we have covered this adequately within the discussion and that expansion of the discussion of Snail overexpression within in vivo conditions beyond what is already present would be excessively speculative. As discussed within the manuscript, EMT partially underlines the malignant phenotype and resistance of PDAC, and high levels of Snail closely correlate with lymph node activation and distant metastasis in humans (Yin et al 2007, Journal of Surgical Research; Hotz et al. 2007, JBC; von Burstin et al.2009, Gastroenterology; Zhou et al. 2016, Cancer Letters; Xu et al. 2015, Nature Communications; Guo et al. 2014, International Journal of Cancer). In vivo models of PDAC have confirmed that Snail overexpressing cells undergo EMT and display invasive behaviours (Yin et al. 2007, Journal of Surgical Research). Although reprogrammed metabolism and increase glycolytic activity are found in a number of cancers that can undergo EMT and Snail levels correlate with both EMT status and increased tumour metastasis in patients, we are not aware of any patient-based work within the literature that confirms the link between Snail and metabolomic reprogramming directly in vivo and thus we must rely primarily upon mechanistic and model studies to interpret our findings.
7. LDH-B protein levels should have shown for comparison with mRNA levels in both cell lines in Figures 2 and 3. Why there is more lactate accumulation in these cells, is this is derived from non-carbohydrate source (gluconeogenesis) or does it have direct correlation with increased glucose transport?
Unfortunately we were unable to perform immunoblotting of LDH-B due to lack of a suitable antibody, nor did we have time to perform any additional experiments given the tight turnaround time for our response.
Our mRNA results do suggest the latter, that lactate accumulation has a direct correlation with increased glucose transport. HPDE Snail overexpressing cells have increased levels of Glut1 (SLC2A1) and Glut3 (SLC2A3) mRNA relative to controls, pointing towards an increased ability to the cells to uptake glucose from the media (Figure 3). 13C flux results also confirmed this, that the increased accumulation of lactate was mostly explained by glycolysis increased by Snail overexpression (i.e., aerobic glycolysis). We have indicated this in the text.
Nonetheless, as lactate was not fully enriched, our results also suggested that non-glucose sources contributed to lactate accumulation (e.g., glutaminolysis). However, this contribution was minor compared to aerobic glycolysis.
8. Increase in Steroyl co-A desaturase 1 (SCD1) levels were observed in many metabolic diseases (diabetes, atherosclerosis, obesity and hypertriglyceridemia) and tumor malignancy. Increase in the expression of steroyl co-A desaturase1 might be due to increased glucose transport into the cellular via glut receptors. Literature studies have shown that the SCD1 is located in the endoplasmic reticulum (ER). Did the authors study any ER stress related markers?
We did not study any ER stress-related markers in the present study and were unable to run such analysis given the time restriction imposed by the journal for our response to this review.
9. Authors might consider performing a comprehensive analysis such as next generation sequencing (RNAseq) and proteomic studies using in vivo (pancreatic cancer tissue from ductal origin) and in vitro studies (Panc1 cells). The data might provide an evidence of cell-type specific genes and proteins that are involved in EMT transitions.
We agree with the reviewer that these analyses would be highly interesting. Such an undertaking is beyond the purview of the present manuscript, and could not be feasibly conducted with resources available to us within the timeline given to use for this response.
Reviewer 3 Report
The authors present interesting findings on the metabolic effects of EMT induced via Snail-overexpression in PDAC. They contrast the effects between two cell-lines that lie on opposite ends of the EMT spectrum. The metabolic characterizations performed are extensive and are largely supportive of the conclusions drawn by the authors. In particular, they observe an increase in glycolytic flux in both lines upon EMT induction. This phenotype was accompanied by reduction in oxidative mitochondrial metabolism as observed from Seahorse assays and flux analysis. Further, most interestingly, the authors did not observe a change in chemoresistance after EMT induction. Although these results are very interesting they oppose some of the findings in the study published in Nature by Zheng et al. (Zheng X, Carstens JL, Kim J, Scheible M, Kaye J, Sugimoto H, Wu CC, LeBleu VS, Kalluri R. Epithelial-to-mesenchymal transition is dispensable for metastasis but induces chemoresistance in pancreatic cancer. Nature. 2015).
There are a few concerns I would like the authors to address prior to publication:
Abstract: "Snail-overexpressed pancreatic cells additionally displayed increased glucose uptake and lactate production with concomitant reductions in mitochondrial electron transport chain proteins." The authors should rephrase the abstract to highlight the key results and make them clearer. Specifically this statement where they start by mentioning uptake/secretion and end with protein expression, even though they find evidence of changes in the oxidative metabolism at the metabolic level.
Experimental Section and Results: 13C flux analysis. The authors should provide a table with flux results and the 13C-MFA model used for this study. This will give the readers a complete picture of the flux analysis results and not just rely on the key observations noted by the authors.
Figure 2C and G. The authors use a wound healing assay to represent the EMT phenotype induced by Snail overexpression. While there is nothing wrong with that approach, it is uncanny that the empty vector controls for tumorigenic Panc-1 line and the non-tumorigenic HPDE lines have the same migration capacity. The authors should clarify this so as to maintain their initial assumption that Panc-1 and HPDE cells not only have differential expression of EMT markers, but also have a different EMT phenotype. Unless, comparing migration capacity of different cell-lines is like comparing apples and oranges, it would be prudent for the authors to address this.
Results, Line 290. The authors mention "glycolytic capacity" as a metabolic marker, however, they should briefly mention the importance of glycolytic capacity for readers without a background in cellular metabolism.
Results, Line 384. The authors do not mention how changes in lipid metabolism fit in with the changes observed in glucose and mitochondrial metabolism. They should at the very least, refer to their discussion section. The section as is breaks the continuity of the results presented before and after.
Minor revisions:
Line 225, reference #46 is not formatted properly
Line 417, Typo: Figure 5A, 5E is referred to as Figure 54A, 45E
Author Response
The authors present interesting findings on the metabolic effects of EMT induced via Snail-overexpression in PDAC. They contrast the effects between two cell-lines that lie on opposite ends of the EMT spectrum. The metabolic characterizations performed are extensive and are largely supportive of the conclusions drawn by the authors. In particular, they observe an increase in glycolytic flux in both lines upon EMT induction. This phenotype was accompanied by reduction in oxidative mitochondrial metabolism as observed from Seahorse assays and flux analysis. Further, most interestingly, the authors did not observe a change in chemoresistance after EMT induction. Although these results are very interesting they oppose some of the findings in the study published in Nature by Zheng et al. (Zheng X, Carstens JL, Kim J, Scheible M, Kaye J, Sugimoto H, Wu CC, LeBleu VS, Kalluri R. Epithelial-to-mesenchymal transition is dispensable for metastasis but induces chemoresistance in pancreatic cancer. Nature. 2015).
There are a few concerns I would like the authors to address prior to publication:
Abstract: "Snail-overexpressed pancreatic cells additionally displayed increased glucose uptake and lactate production with concomitant reductions in mitochondrial electron transport chain proteins." The authors should rephrase the abstract to highlight the key results and make them clearer. Specifically this statement where they start by mentioning uptake/secretion and end with protein expression, even though they find evidence of changes in the oxidative metabolism at the metabolic level.
We have amended the abstract to include additional findings within the manuscript, including impact on oxidative metabolism (as well as electron transport chain proteins).
Experimental Section and Results: 13C flux analysis. The authors should provide a table with flux results and the 13C-MFA model used for this study. This will give the readers a complete picture of the flux analysis results and not just rely on the key observations noted by the authors.
We have prepared an additional table to be included as Supplementary information that details the results of the 13C metabolic flux analysis in this study.
Figure 2C and G. The authors use a wound healing assay to represent the EMT phenotype induced by Snail overexpression. While there is nothing wrong with that approach, it is uncanny that the empty vector controls for tumorigenic Panc-1 line and the non-tumorigenic HPDE lines have the same migration capacity. The authors should clarify this so as to maintain their initial assumption that Panc-1 and HPDE cells not only have differential expression of EMT markers, but also have a different EMT phenotype. Unless, comparing migration capacity of different cell-lines is like comparing apples and oranges, it would be prudent for the authors to address this.
Comparing migratory capacity between the two cell lines is indeed like comparing apples with oranges. These experiments were performed at different times under different experimental conditions (i.e. culture media), and so directly comparing between the two data in this instance is not appropriate.
Results, Line 290. The authors mention "glycolytic capacity" as a metabolic marker, however, they should briefly mention the importance of glycolytic capacity for readers without a background in cellular metabolism.
We have updated this section of the text to better explain both maximal glycolysis and glycolytic capacity.
Results, Line 384. The authors do not mention how changes in lipid metabolism fit in with the changes observed in glucose and mitochondrial metabolism. They should at the very least, refer to their discussion section. The section as is breaks the continuity of the results presented before and after.
Although changes in lipid metabolism are a noted metabolic adaptation of many cancers, the reviewer makes an excellent point regarding the lipid metabolism data in the current manuscript and the disruption of the continuity of the results section. Given that additional experiments (e.g. lipidomics) would be required to determine the potential consequences of the changes SCD1 expression, we have decided to remove the impact of snail overexpression on lipid metabolism findings from the manuscript. The removal of these results has no impact on the major conclusions of the study, but does improve the overall flow of the manuscript.
Minor revisions:
Line 225, reference #46 is not formatted properly
This has been corrected.
Line 417, Typo: Figure 5A, 5E is referred to as Figure 54A, 45E
This has been corrected.
Reviewer 4 Report
Liu et al. investigated the role of epithelial-mesenchymal transition-inducing transcription factor (EMT-TF), Snail, in pancreatic cancer cells by focusing their interest on EMT and metabolism. In recent years, there is evidence in the literature that the transcription factor for TEM also plays a role in the regulation of glucose metabolism enzymes.
This second version (revised) of the study is clear and well conducted.
Author Response
We thank the reviewer for their comments, which have helped to improve the manuscript.
Round 2
Reviewer 1 Report
The authors provided some new data, improved figures and clarified several aspects of the manuscript. Thus it is now a substantially improved manuscript suitable for publication in JCM.
Author Response

(The authors gave the same response as above.)
